# Blind Watermarking for Hiding Color Images in Color Images with Super-Resolution Enhancement

**DOI:** 10.3390/s23010370

**Published:** 2022-12-29

**Authors:** Hwai-Tsu Hu, Ling-Yuan Hsu, Shyi-Tsong Wu

**Affiliations:** 1Department of Electronic Engineering, National I-Lan University, Yilan 26047, Taiwan; 2Department of Information Management, St. Mary’s Junior College of Medicine, Nursing and Management, Yilan 26647, Taiwan

**Keywords:** blind color image watermarking, discrete cosine transform, magnitude gap adjustment, super-resolution, generative adversarial network

## Abstract

This paper presents a novel approach for directly hiding the pixel values of a small color watermark in a carrier color image. Watermark embedding is achieved by modulating the gap of paired coefficient magnitudes in the discrete cosine transform domain according to the intended pixel value, and watermark extraction is the process of regaining and regulating the gap distance back to the intensity value. In a comparison study of robustness against commonly encountered attacks, the proposed scheme outperformed seven watermarking schemes in terms of zero-normalized cross-correlation (ZNCC). To render a better visual rendition of the recovered color watermark, a generative adversarial network (GAN) was introduced to perform image denoising and super-resolution reconstruction. Except for JPEG compression attacks, the proposed scheme generally resulted in ZNCCs higher than 0.65. The employed GAN contributed to a noticeable improvement in perceptual quality, which is also manifested as high-level ZNCCs of no less than 0.78.

## 1. Introduction

Nowadays, multimedia data are stored in a digital form suitable for access and distribution on the Internet. Among the available types of multimedia (e.g., text, images, audio, and video), color images are the most frequently used data on openly accessible network platforms [1,2]. While an extensive amount of image data is continuously transmitted over networks, illicit usage of color images has resulted in problems such as unauthorized or unlicensed copying, intellectual property infringements, and malicious tampering. The issue of copyright violation has drawn considerable attention from researchers and content providers who continuously seek feasible countermeasures to protect their ownership and copyright material. To this end, digital watermarking is an effective means of securing data content [3,4,5]. It plays a pivotal role in information security and has broad application prospects.

Image watermarking is the process of embedding proprietary information into carrier images via the use of specific data processing algorithms. The embedded information (or simply a watermark) is supposedly imperceptible and sufficiently robust to withstand unintentional modifications and/or intentional attacks. In addition, the watermark information should be retrievable and easy to verify whether the inspected image has tampering or infringement problems.

Imperceptibility, robustness, and capacity are three mutually conflicting properties of image watermarking [6]. The balance between these often relies on the adjustment of the embedding strength of the watermark. Given that the payload capacity is fixed, increasing the embedding strength may improve the robustness but impair the imperceptibility. To date, it has remained a challenge to find an ideal watermarking scheme that can comprehensively address the three requirements.

Image watermarking can be divided into blind and non-blind categories. Those belonging to the non-blind category require the original image and/or additional information for watermark extraction, whereas blind ones require neither. As the original image is not always accessible during the stage of watermark extraction, the blind approach is most prevalent in practical applications. Image watermarking schemes can also be classified based on the domain chosen to hide the watermark. Transform-domain approaches are widely used as they can take advantage of an image’s intrinsic characteristics and human visual perception. Commonly observed transformations for image watermarking comprise the discrete Fourier transform (DFT) [7,8,9], discrete wavelet transform (DWT) [10,11,12,13], discrete cosine transform (DCT) [14,15,16,17,18], and various matrix decompositions (MD) [19,20,21,22,23,24]. Combinations of the abovementioned transforms were also attempted with encouraging results [25,26,27,28,29].

Although watermarks can be any form of digital data conveying ownership information, the most frequently found watermarks in the literature were image logos. This can be attributed to the fact that the performance of an image watermarking scheme can be easily identified through visual inspection of the recovered watermarks. Our literature survey indicates that there has been no shortage of high-capacity color watermarking schemes proposed in recent years. The following are some examples. In [9], Zhang et al. exploited the unique features of the direct current (DC) component of the DFT. Each binary bit was determined by manipulating the inequality relationship between the DC components of two adjacent image blocks of size 2×2. In the original literature, the authors hid two 24-bit color images of 32 × 32 pixels in a carrier color image of size 512 × 512, but it is possible to embed 64×64×3×8 bits into the host image if every 2×2 block is used to perform watermarking.

After taking the Haar transform on each image block, Liu et al. [30] chose the coefficient with the largest energy to perform quantization. Given that one bit was hidden in every two 2×2 blocks, the maximum allowable capacity was the exact number of 64×64×3×8 bits. As a result, the proposed watermarking scheme achieved a payload capacity of 3/8 (=64×64×24/512×512) bit per pixel (bpp).

The concept of embedding watermark bits into the transformed coefficients derivable from another domain was also found in [27,31]. In [27], Su et al. designed a combined-domain watermarking scheme that nominally modulated the first-level approximation coefficients of the DWT. An essential feature of this scheme lies in the acquisition of the targeted DWT coefficient in the spatial domain. Such a design exploits not only the computational efficiency of spatial-domain watermarking schemes but also the strong robustness of transform-domain schemes. Yuan et al. adopted a similar strategy [31], which explores the DC component of a 2D DCT in the spatial domain to achieve large-capacity watermarking. Here, a binary bit is characterized by the relativity of the DC coefficients between adjacent blocks. For these two aforementioned watermarking schemes, the resultant payload capacity can reach as high as 3/8 bpp as long as the transformed coefficients are derived from block matrixes of 2×2 pixels.

Following the division of the carrier image into block matrices, Chen et al. [32] presented an efficient blind watermarking algorithm using the Walsh–Hadamard transform (WHT). A binary bit can be embedded into a WHT matrix block of size 4 × 4 by fine-tuning the paired coefficients in the first row of the transformed matrix. Regarding the class of MD-based image watermarking, Hsu et al. [23] proposed a high-capacity QR decomposition (QRD)-based image watermarking algorithm that manipulates two pairs of elements drawn from the first column of the orthogonal matrix after applying the QRD to a block of size 4×4 pixels. Meanwhile, Liu et al. [24] proposed embedding watermark information into the maximum eigenvalue of the Schur decomposition of selected matrices using quaternary coding. This scheme claimed to be efficient because multiple bits were accommodated in the quaternary code. As these forgoing three approaches proceeded from block matrixes of size 4×4, a high capacity at a level of 3/8 bpp is feasible if each block can contain 2 bits.

Although the aforementioned high-capacity blind color image watermarking schemes have achieved certain degrees of success, their robustness against severe attacks could not yet match acceptable expectations. Hence, this study aims to develop a scheme capable of performing efficient high-capacity blind color image watermarking, along with the use of a generative adversarial network (GAN) to enhance the visual quality of extracted watermarks. Consequently, the contributions in this study include two aspects: (1) the development of a novel blind watermarking scheme that enables a direct and effective embedding of 8-bit unsigned pixel values, which facilitates the goal of hiding color images in color images; (2) the incorporation of a GAN to enhance the watermark images retrieved from the watermarked carrier images, thus making the watermarks more visually recognizable.

The remainder of this paper is organized as follows. Section 2 outlines the technical background pertaining to the fundamentals of DCT and super-resolution (SR). Section 3 elucidates the procedures for watermark embedding and extraction in the DCT domain. Section 4 presents experimental results. Improvements owing to the incorporation of the GAN are also reported. Finally, conclusions are presented in Section 5.

## 2. Preliminaries

### 2.1. DCT-Based Watermarking

Among the transformations used in digital image processing, the DCT has drawn considerable attention as it can efficiently compact the signal energy at low frequencies [33]. This property makes the DCT highly suitable for applications in image compression and watermarking. The two-dimensional (2D) DCT performed on an image matrix acts the same as a one-dimensional (1D) DCT performed along a single dimension, followed by another 1D DCT in the other dimension. The 2D DCT for an image block matrix $X={\left\{X_{mn}\right\}}_{M\times N}$ and output DCT matrix $Y={\left\{Y_{pq}\right\}}_{M\times N}$ is defined as
(1)$$Y_{pq}=\alpha_{p}\alpha_{q}\sum_{m=0}^{M-1}\sum_{n=0}^{N-1}X_{mn}\cos\frac{\pi (2m+1)p}{2M}\cos\frac{\pi (2n+1)q}{2N},\begin{array}{c} 0\leq p\leq M-1\\ 0\leq q\leq N-1 \end{array}$$
with
αp=1M,p=0;2M,1≤p≤M−1  αq=1N,p=0;2N,1≤p≤N−1.  
where $X_{mn}$ denotes the ${\left(m,n\right)}^{th}$ pixel value of the image block, $Y_{pq}$ represents the frequency component at position (p,q), and M and N correspond to the row and column size of X, respectively. After the 2D DCT, the output matrix Y containing direct-current (DC) and alternating-current (AC) components can be obtained. The DC coefficient situated at the top-left corner of the resulting DCT matrix signifies the average intensity of the image block. Depending on the distance from the DC component, the remaining coefficients in Y can be classified as low-, middle-, and high-frequency ranges.

DCT-based image watermarking is referred to as the modification of DCT coefficients according to specific rules imposed by watermarking. Once the watermarking process is complete, the application of the inverse DCT (IDCT) can recover the image block with concealed watermark information. The formula for the 2D IDCT is expressed as follows:(2)$$X_{mn}=\sum_{p=0}^{M-1}\sum_{q=0}^{N-1}\alpha_{p}\alpha_{q}Y_{pq}\cos\frac{\pi (2m+1)p}{2M}\cos\frac{\pi (2n+1)q}{2N},\begin{array}{c} 0\leq m\leq M-1\\ 0\leq n\leq N-1. \end{array}$$

### 2.2. Super-Resolution Reconstruction

SR is a technique used to reproduce images with higher resolution. This technique can be employed as a post-process to enhance the watermark image. In this study, we only consider single-image SR (SISR) that attempts to reconstruct high resolution from a single low-resolution watermark image. SISR is challenging because high-frequency details are already lost in the low-resolution image. Owing to the advancement of deep learning, reestablishing the complex mapping between low- and high-resolution images is possible using very deep neural networks.

As demonstrated in [34,35,36,37], several methods developed based on convolutional networks have shown promising results. Ledig et al. [38] not only tackled the SISR problem using a deep residual network (ResNet) but also introduced a GAN framework to acquire photo-realistic natural images. Figure 1 illustrates the GAN structure for carrying out SISR. The generator is a 16-block deep ResNet, where the function is denoted by $G_{\theta_{G}}(\cdot )$, capable of recovering a high-resolution image $I_{HR}^{*}$ ($≜G_{\theta_{G}}(I_{LR})$) from a low-resolution image $I_{LR}$. The discriminator, which is expressed as $D_{\theta_{D}}(\cdot )$, renders the probability whether the input is from either the reference or generator distributions. Subsequently, the GAN-based network is optimized to solve an adversarial min-max problem as follows:(3)$$\underset{\theta_{G}}{\min}\underset{\theta_{D}}{\max}E_{I_{HR}∼P_{ref}(I_{HR})}\left[\log D_{\theta_{D}}(I_{HR})\right]+E_{I_{HR}∼P_{G}(I_{LR})}\left[\log\left(1-D_{\theta_{D}}\left(G_{\theta_{G}}(I_{LR})\right)\right)\right],$$
where $\theta_{G}$ and $\theta_{D}$ correspond to the model parameters associated with the generator and discriminator, respectively. Further, E[⋅] denotes the expectation operator based on a probability distribution, and P(⋅) is a probability density function. The basic idea behind the above formulation is to train the generative model $G_{\theta_{G}}(\cdot )$ to fool the discriminator $D_{\theta_{D}}(\cdot )$, which is supposedly trained to distinguish between model-generated and real images. It was reported in [38] that, with the GAN-based framework, using perceptual loss in the discriminator model contributed to significant improvements in perceptual quality.

## 3. Proposed Watermarking Scheme

Figure 2 depicts the entire process of watermark embedding and extraction. The watermarks used in this study were downsized color images of 64×64 pixels. To ensure information security, each watermark image was scrambled using the Arnold transform [39] before embedding. When extracting the watermark, one needs a correct key to descramble the watermark to restore the actual content.

### 3.1. Watermark Embedding via the DCT-MGA Scheme

The proposed DCT-based magnitude gap adjustment (DCT-MGA) scheme starts by partitioning the carrier image into non-overlapping blocks with a size of 8×8. The next step is to take the 2D DCT of each block in the R, G, and B channels individually. After obtaining the required DCT coefficients, watermark embedding is a joint operation of two involved coefficients such that the gap between the two coefficient magnitudes matches the desired length.

Let $Y_{p_{1},q_{1},r_{1}}$ and $Y_{p_{2},q_{2},r_{2}}$ denote the two DCT coefficients that participate in the embedding process. Further, $\left(p_{1},q_{1}\right)$ and $\left(p_{2},q_{2}\right)$ represent the coordinates of the paired coefficients, while $r_{1}$ and $r_{2}$ signify the channel sources for each respective coefficient. For example, $(p_{1},q_{1},r_{1})=$(3,0,1) and $(p_{2},q_{2},r_{2})=$(3,1,1) were selected to embed the red color pixel values. Watermark embedding involves the manipulation of two coefficient magnitudes, namely, $\left|Y_{p_{1},q_{1},r_{1}}\right|$ and $\left|Y_{p_{2},q_{2},r_{2}}\right|$, such that the gap between them can reflect the intended pixel value. For this purpose, we first define the following parameters:(4)$$m_{low}=\min\left\{\left|Y_{p_{1},q_{1},r_{1}}\right|,\;\left|Y_{p_{2},q_{2},r_{2}}\right|\right\},$$
(5)$$m_{high}=\max\left\{\left|Y_{p_{1},q_{1},r_{1}}\right|,\;\left|Y_{p_{2},q_{2},r_{2}}\right|\right\},$$
(6)$$\rho =\frac{\hat{w}(i,j,k)-127.5}{255}\alpha ,$$
where $m_{low}$ and $m_{high}$ are the lower and higher levels of the two magnitudes, respectively, and $\hat{w}(i,j,k)$ denotes the ${\left(i,j\right)}^{th}$ pixel value of the scrambled watermark in the $k^{th}$ channel. A value of 255 in the denominator of Equation (6) is the maximum allowable level of an 8-bit unsigned integer; thus, the midline of the dynamic range becomes 127.5. Further, α signifies the embedding strength. Consequently, the gap (symbolized as ρ) reflects a scaled quantity deviating from the midline, with a distance proportional to $\hat{w}(i,j,k)$. After determining the gap ρ, the two magnitudes are adjusted accordingly.
(7)$${\hat{m}}_{low}=\max\left\{0,\frac{\left|Y_{p_{1},q_{1},r_{1}}\right|+\;\left|Y_{p_{2},q_{2},r_{2}}\right|}{2}-\frac{\rho }{2}\right\}$$
(8)$${\hat{m}}_{high}={\hat{m}}_{low}+\rho$$

Equation (7) adjusts $m_{low}$ in the least-squares sense, but the resulting output ${\hat{m}}_{low}$ must be positive to conform with the definition of magnitude. Equation (8) confines the gap between ${\hat{m}}_{high}$ and ${\hat{m}}_{low}$ to exactly ρ. After determining ${\hat{m}}_{high}$ and ${\hat{m}}_{low}$, we reassign the two involved coefficients as:(9)$$\begin{array}{l} \mathrm{If}\;\left|Y_{p_{1},q_{1},r_{1}}\right|\leq \;\left|Y_{p_{2},q_{2},r_{2}}\right|\\ \;\;\;{\hat{Y}}_{p_{1},q_{1},r_{1}}=\mathrm{sgn}\left(Y_{p_{1},q_{1},r_{1}}\right)\cdot {\hat{m}}_{low};\\ \;\;\;{\hat{Y}}_{p_{2},q_{2},r_{2}}=\mathrm{sgn}\left(Y_{p_{2},q_{2},r_{2}}\right)\cdot {\hat{m}}_{high};\\ \mathrm{else}\\ \;\;\;{\hat{Y}}_{p_{1},q_{1},r_{1}}=\mathrm{sgn}\left(Y_{p_{1},q_{1},r_{1}}\right)\cdot {\hat{m}}_{high};\\ \;\;\;{\hat{Y}}_{p_{2},q_{2},r_{2}}=\mathrm{sgn}\left(Y_{p_{2},q_{2},r_{2}}\right)\cdot {\hat{m}}_{low}, \end{array}$$
where sgn(⋅) denotes a sign function. This embedding process is repeated for every block and channel involved. After replacing the modified coefficients in the DCT block matrix, we used the IDCT to obtain the watermarked image block.

In this study, we deliberately chose three pairs of adjacent DCT coefficients to embed three pixel values in 8-bit unsigned integer format. While watermarks hidden at high frequencies are vulnerable to high-frequency attacks (e.g., low-pass filtering and image compression), watermarks at low frequencies cannot resist attacks such as unsharp filtering and histogram equalization. A compromise can be reached by selecting DCT coefficients situated in the middle frequencies. Figure 3 illustrates the locations of the three coefficient pairs selected to embed the three pixel values in this study. Specifically, the three designated pairs are $\left\{\left(3,0,1),\;(3,1,1\right)\right\}$, $\left\{\left(2,1,2),\;(2,2,2\right)\right\}$, and $\left\{\left(1,2,3),\;(0,3,3\right)\right\}$, where the first two indices in each coordinate representation indicate the location of the 2D DCT matrix. The third index signifies the color channels with 1, 2, and 3 corresponding to the red, green, and blue colors, respectively. The motives for these pair arrangements are twofold. First, the watermarking process generally requires an increase in one magnitude but a decrease in the other for each pair. As the energy of each color component remains roughly balanced, modifications of a pair of DCT coefficients in adjacent locations with the same color are expected to yield less variation in color intensity. Second, because textual patterns of adjacent DCT coefficients retain partial similarity, the image reconstructed after watermarking can show more structural similarities.

### 3.2. Watermark Extraction and Regulation

Watermark extraction is the process of retrieving a watermark from a watermarked color image. The right half of Figure 2 shows the extraction procedure. The first two steps involve partitioning the watermarked color image into non-overlapping blocks of size 8×8 and then applying 2D DCT to these blocks. For each set of $\left\{{\tilde{Y}}_{pqr}\right\}$ derived from an image block, we select three pairs of DCT coefficients (as shown in Figure 3) and then compute the pixel value using the following formula:(10)$$\tilde{w}(i,j,k)=\frac{255}{\alpha }\left(\left|{\tilde{Y}}_{p_{1},q_{1},r_{1}}\right|-\left|{\tilde{Y}}_{p_{2},q_{2},r_{2}}\right|\right)+127.5,\;k=1,2,3,$$
where $\tilde{w}(i,j,k)$ denotes the retrieved ${\left(i,j\right)}^{th}$ pixel value in the $k^{th}$ channel and the tilde implies that the watermark image may have suffered from attacks.

In our experiments, it was observed that, under certain attacks, the extracted watermarks might appear brighter or darker than the original watermarks. Interestingly, watermarks with altered luminance are still visually recognizable, suggesting that the attacks have merely caused the pixel values to be rescaled. Restoring the retrieved pixel values back to the normal range is expected to provide a better view of the watermark. Consequently, a simple remedial algorithm is introduced below to rectify the recovered watermark.

According to Equation (10), the output is obtained by adding a deviation term (i.e., $d≜\frac{255}{\alpha }\left(\left|{\tilde{Y}}_{p_{1},q_{1},r_{1}}\right|-\left|{\tilde{Y}}_{p_{2},q_{2},r_{2}}\right|\right)$) to the midline value 127.5. Because $\tilde{w}(i,j,k)$ can only fall between 0 and 255, the absolute value of the deviation term (i.e., $\left|d\right|$) must not exceed 127.5. Luminance regulation begins with the sorting of $\left|d\right|$’s gathered from every channel. Then, the average in the range 90–95% of the sorted $\left|d\right|$’s, termed η, serves as a gauge to mark the upper margin. Depending on the resulting η, the deviation term is proportionally restrained to a range between $\psi_{low}$ and $\psi_{up}$ as follows:(11)$${\tilde{w}}_{reg}(i,j,k)=\left\{\begin{array}{cc} \left(\tilde{w}(i,j,k)-127.5\right)\times \psi_{up}/\eta +127.5, & \eta >\psi_{up};\\ \left(\tilde{w}(i,j,k)-127.5\right)\times \psi_{low}/\eta +127.5, & \eta <\psi_{low};\\ \tilde{w}(i,j,k), & \mathrm{otherwise}. \end{array}\right.$$
where ${\tilde{w}}_{reg}(i,j,k)$ denotes the regulated outcome from the initially retrieved pixel value $\tilde{w}(i,j,k)$. Terms $\psi_{up}$ and $\psi_{low}$ respectively represent the upper and lower boundaries of the expected η. These two boundaries can be deduced from the dataset of possible watermarks. In this manner, the regulator restrains the luminance of the color watermark to a normal value. Note that the scale adjustment (as given in Equation (11)) has no influence on the textural content; however, it makes the watermark image more visually distinguishable. After gathering all pixels $\left\{{\tilde{w}}_{reg}(i,j,k)\right\}$, the last step of watermark extraction is to use the decryption key to discern the watermark image.

## 4. Performance Evaluation

The test materials comprise 16 color images of 512×512 pixels collected from the USC–SIPI [40] and CVG–UGR [41] image databases. Figure 4 shows the 16 color images in a 4×4 array. The watermarks were obtained by downsampling the 16 color images to a size of 64 × 64 pixels. To accommodate such a watermark in a normal-size carrier image, each 8×8-sized image block must contain three 8-bit unsigned integer values. The payload capacity was thereby 3/8 $\left(=\left(64\times 64\times 3\times 8\right)/(512\times 512)\right)$ bpp. Consequently, there are 256 possible combinations between the carrier images and intended watermarks. To enhance the security of the watermark information, each color channel of the color watermark image was scrambled before embedding using the Arnold transform [39]. To this end, the retrieved watermark must undergo a descrambling process after watermark extraction. Figure 5 presents the test watermarks with their scrambled versions examined in our experiments.

To render a balanced performance between robustness and imperceptibility, this study set the embedding strength α (as defined in Equation (6)) at 90 for the proposed DCT-MGA. The two boundaries for luminance regulation were chosen as $\psi_{up}=83$ and $\psi_{low}=120$, which correspond to the mean value plus/minus the standard deviation of the η’s derived from all the images in the IAPR TC-12 database [42]. In addition to the proposed DCT-MGA, seven recently published blind color image watermarking schemes were considered for performance evaluation and comparison. For simplicity, the seven schemes are abbreviated hereinafter as QRMM22 [23], WHT21 [32], Haar21 [30], Schur21 [24], DFT20 [9], DCT20 [31], and DWT20 [27], with the last two digits signifying the publication year.

### 4.1. Imperceptibility Test

When determining the embedding strength, trade-offs exist amid imperceptibility, robustness, and capacity. In general, a weak embedding strength is conducive to imperceptibility but detrimental to robustness. Raising the embedding strength can reinforce the watermark hidden in the carrier image but may also impair the visual quality of the watermarked image. To probe the influence of embedding strength, we conducted a pilot experiment to determine an adequate embedding strength that achieves balance in imperceptibility and robustness.

To assess the distortion of the carrier image due to the watermarking process, we adopted the peak signal-to-noise ratio (PSNR) and mean structural similarity (mSSIM) metrics given below to assess the degradation of the image quality.
(12)$$PSNR(I,\;\hat{I})=10\times \log_{10}\left(\frac{255^{2}}{\frac{1}{M_{row}\times N_{col}\times 3}\sum_{t=1}^{3}\sum_{m=1}^{M_{row}}\sum_{n=1}^{N_{col}}{\left(I_{m,n,t}-{\hat{I}}_{m,n,t}\right)}^{2}}\right);$$
(13)$$mSSIM(I,\;\hat{I})=\frac{1}{L\times K\times 3}\sum_{t=1}^{3}\sum_{l=1}^{L}\sum_{k=1}^{K}SSIM\left(B_{l,k,t},{\hat{B}}_{l,k,t}\right),$$
where $I={\left\{I_{m,n,t}\right\}}_{N_{row}\times N_{col}\times 3}$ and $\hat{I}={\left\{{\hat{I}}_{m,n,t}\right\}}_{N_{row}\times N_{col}\times 3}$, respectively, represent the original and watermarked images of $M_{row}\times N_{col}$ pixels with a depth of three color channels. $B_{l,k,t}$ and ${\hat{B}}_{l,k,t}$ in Equation (13) correspond to the ${\left(l,k\right)}^{th}$ windows in the $t^{th}$ channel acquired from I and $\hat{I}$, respectively. The denominator value L×K×3 indicates the number of image blocks. The function SSIM(⋅) is responsible for measuring the degree of similarity between $B_{r,s,t}$ and ${\hat{B}}_{r,s,t}$.
(14)$$\mathrm{SSIM}\left(B_{l,k,t},{\hat{B}}_{l,k,t}\right)=\frac{\left(2\mu_{B_{l,k,t}}^{}\mu_{{\hat{B}}_{l,k,t}}^{}+k_{1}\right)\left(2\sigma_{B_{l,k,t}{\hat{B}}_{l,k,t}}^{}+k_{2}\right)}{\left(\mu_{B_{l,k,t}}^{2}+\mu_{{\hat{B}}_{l,k,t}}^{2}+k_{1}\right)\left(\sigma_{B_{l,k,t}}^{2}+\sigma_{{\hat{B}}_{l,k,t}}^{2}+k_{2}\right)},$$
where $\mu_{B_{l,k,t}}^{}$ and $\sigma_{B_{l,k,t}}^{2}$, respectively, represent the mean and variance of $B_{l,k,t}$; $\sigma_{B_{l,k,t}{\hat{B}}_{l,k,t}}^{}$ denotes the covariance between $B_{l,k,t}$ and ${\hat{B}}_{l,k,t}$; $k_{1}$ and $k_{2}$ are values introduced to ensure the stability of SSIM. By default, these values are $k_{1}=0.01$ and $k_{2}=0.02$.

Figure 6 depicts the average PSNRs and mSSIMs for α in the range 60–100 with increments of 2. Based on the experimental results shown in Figure 6, we chose α as 90 as this choice led to a PSNR near 38.5 dB and an mSSIM of approximately 0.96. In general, watermarked images with this PSNR and mSSIM are deemed to possess satisfactory quality.

Note that the seven watermarking schemes under comparison were intended to hide binary information, but not all of them were initially designed to attain a payload capacity as high as 3/8 bpp. If such a high capacity is under request, the original embedding strength specified in the literature may lead to unacceptable PSNR and mSSIM outcomes. Consequently, while implementing the compared schemes, we had to adjust the parameters relevant to the embedding strength such that the PSNRs and mSSIMs fell within an acceptable range.

Table 1 presents the statistical results measured for every possible combination of test images and watermarks. With the use of the specified embedding strength, the average PSNR and mSSIM obtained from the proposed DCT-MGA were 38.48 dB and 0.959, respectively, both of which ranked the highest among all compared schemes. The PSNRs acquired by DCT20 and DFT20 fell below 33 dB, suggesting that these two schemes may cause excessive pixel modification. Ironically, the mSSIMs resulting from these two schemes were not the worst. This can be attributed to the fact that watermark embedding merely affects the DC component, which causes less damage to the textural structure of the carrier image. The worst mSSIM value generally occurred in the case of DWT20. Despite the average PSNR obtained from DWT20 remained above 36 dB, the resulting mSSIM unexpectedly dropped to 0.925. This is thought to be due to the watermark being embedded in a relatively wide frequency sub-band.

### 4.2. Robustness Test

The second phase of the performance evaluation focused on the robustness of the compared watermarking schemes in the presence of commonly encountered attacks. The types of attacks considered in this study included image compression, noise corruption, filtering, histogram equalization, geometric correction, cropping, and luminance adjustment. Table 2 lists the specifications for the intended attacks.

As discussed in Section 3, the proposed DCT-MGA directly embeds 8-bit unsigned pixel values into the specific DCT coefficients. By contrast, all other compared watermarking schemes can only embed binary bits. Because the embedding data type of the proposed DCT-MGA differs from the others that perform binary watermarking, such a difference imposes difficulties in performance comparison. One feasible assessment metric applicable to either type of watermarking is zero-normalized cross-correlation (ZNCC), which characterizes the similarity of two data sequences without concerning the data type. For a watermark constituted by pixel values, ZNCC is defined as
(15)$$\begin{array}{l} ZNCC(W_{v},{\tilde{W}}_{v})=\\ \frac{\sum_{p=1}^{n_{L}}\sum_{q=1}^{n_{K}}\sum_{r=1}^{n_{ch}}\left(w_{v}(p,q,r)-{\bar{w}}_{v}\right)\times \left({\tilde{w}}_{v}(p,q,r)-{\bar{\tilde{w}}}_{v}\right)}{\sqrt{\left(\sum_{p=1}^{n_{L}}\sum_{q=1}^{n_{K}}\sum_{r=1}^{n_{ch}}{\left(w_{v}(p,q,r)-{\bar{w}}_{v}\right)}^{2}\right)\times \left(\sum_{p=1}^{n_{L}}\sum_{q=1}^{n_{K}}\sum_{r=1}^{n_{ch}}{\left({\tilde{w}}_{v}(p,q,r)-{\bar{\tilde{w}}}_{v}\right)}^{2}\right)}} \end{array}$$
where ${\bar{w}}_{v}$ denotes the mean of the pixel values obtained from a watermark image $W_{v}$= $\left\{w_{v}\right\}$ of size $n_{L}\times n_{K}$(=64×64) with $n_{ch}$(=3) channels, and the tilde signifies retrieval after a possible attack.

Alternatively, the ZNCC for the watermark composed of binary bits is
(16)$$\begin{array}{l} ZNCC(W_{b},{\tilde{W}}_{b})=\\ \frac{\sum_{p=1}^{n_{L}}\sum_{q=1}^{n_{K}}\sum_{r=1}^{n_{ch}}\sum_{s=1}^{n_{b}}\left(w_{b}(p,q,r,s)-{\bar{w}}_{b}\right)\times \left({\tilde{w}}_{b}(p,q,r,s)-{\bar{\tilde{w}}}_{b}\right)}{\sqrt{\left(\sum_{p=1}^{n_{L}}\sum_{q=1}^{n_{K}}\sum_{r=1}^{n_{ch}}\sum_{s=1}^{n_{b}}{\left(w_{b}(p,q,r,s)-{\bar{w}}_{b}\right)}^{2}\right)\times \left(\sum_{p=1}^{n_{L}}\sum_{q=1}^{n_{K}}\sum_{r=1}^{n_{ch}}\sum_{s=1}^{n_{b}}{\left({\tilde{w}}_{b}(p,q,r,s)-{\bar{\tilde{w}}}_{b}\right)}^{2}\right)}} \end{array}$$
where ${\bar{w}}_{b}$ denotes the mean of the binary values obtained from a watermark image of $n_{L}\times n_{K}\times n_{ch}\times n_{b}$ bits. Here, $n_{L}=n_{K}=64$, $n_{ch}=3$ and $n_{b}=8$.

As presented in Table 3, the proposed DCT-MGA rendered satisfactory ZNCCs that were normally larger than 0.65, except for JPEG compression. The lowest ZNCC occurred in Case A.2, where the quality factor was set to 40. Nonetheless, the seemingly disappointing score of 0.315 was still the highest among the eight compared schemes. In addition, except for cases D (salt-and-pepper noise corruption) and J (histogram equalization), the DCT-MGA generally obtained the highest rank. Other schemes may perform well, but they cannot handle all the types of attacks considered in this study. In Table 4, two typical watermarks, retrieved from the watermarked “Lena” color image for all the different schemes in the presence of various attacks, are presented as extracted watermark examples. A simple visual inspection of these watermarks can easily demonstrate the superiority of the proposed DCT-MGA.

### 4.3. Watermark Enhancement

Because the recovered watermark contains only 64×64 pixels, such a size makes it blurry when zoomed in. A possible method to improve the quality of the image watermark is to enhance the clarity and resolution using deep learning techniques. In this study, we adopted the SR-GAN framework developed in [38] to perform image denoising and SR reconstruction after recovering watermarks from the watermarked images. The employed generator utilizes a residual network (ResNet) capable of converting low-resolution images (64×64 in size) to high-resolution images (of size 256×256). SR-ResNet can operate independently to improve the visual quality of the watermark; however, the output images are often overly smoothed [38,43]. With the exploitation of the GAN, the generator is trained to fool the discriminator rather than to minimize the difference between the desired and generated image outputs. It was reported in [38] that, with the adoption of a perceptual loss function, the SR-GAN is empowered to recover photo-realistic textures from downsampled images.

The watermarks used for training the GAN comprise 1000 color images taken from the IAPR TC-12 database [42]. The four images (i.e., “Lena,” “Baboon,” “Peppers,” and “F16”) shown on the top row of Figure 4 served as the carrier images in a simulation of watermark extraction under attacks. Prior to network training, we adopted a preprocessing method similar to image augmentation to expand the number of image samples. In addition to the ideal case without any attack, we applied every attack listed in Table 2 to the four watermarked images and recovered the watermarks using Equation (10). The descrambled watermark (of size 64×64) and its original high-resolution image (of size 256×256) form an input—output sample pair. Overall, 76,000 (4×19×1000) watermark samples were used in the GAN training.

The improvement due to the SR-GAN is demonstrated in Figure 7, which presents an exemplary watermark retrieved from the watermarked “Lena” image under various attacks. Each subplot in Figure 7 comprises three parts: the left one shows the retrieved watermark on the upper-left corner and its up-sampled version; the middle one presents the high-resolution image resulting from the SR-ResNet trained based on the mean squared error (MSE) between the recovered high-resolution image and the ground truth; the right one corresponds to the so-called “photo-realistic” images rendered by the SR-GAN. As evidenced by this demonstration, SR-ResNet inherits the merits of convolutional neural networks, which not only enhance the resolution but also suppress the noise caused by attacks. Nonetheless, the use of MSE as a loss function in the SR-ResNet tends to yield overly smooth textures. With the involvement of the GAN, the generator network can render perceptually satisfying images that match the expected resolution.

As also revealed in Figure 7, the effect due to JPEG compression deserves special attention. It can be found in Subplots A.1 and A.2 that the extracted watermarks were not only noisier but also had chromatic aberrations in the associated colors. Such a phenomenon can be attributed to the poor preservation of the chroma information in the high-frequency region. Thanks to the SR-ResNet and SR-GAN, the problems with the noise, chroma, and limited resolution could be substantially ameliorated.

To determine whether SR-ResNet and SR-GAN are conducive to the recognition of the extracted watermark, we computed the ZNCCs between the recovered and real watermarks in the presence of all the attacks considered in this study. Figure 8 presents the results as a bar chart, wherein each attack category comprises four bins, denoted as “w64,” “BIw256,” “SR-ResNetw256,” and “SR-GANw256.” Specifically, “w64” indicates the results derived from watermarks of size 64×64, while “BIw256” corresponds to the results from the bicubically interpolated watermarks of size 256 × 256. Meanwhile, “SR-ResNetw256” and “SR-GANw256” represent those obtained from the SR-ResNet and SR-GAN, respectively. Several conclusions can be drawn from Figure 8. First, the interpolated watermarks did not improve the ZNCCs. Second, SR-ResNet substantially enhanced the ZNCC when the watermarks were severely damaged by malicious attacks. This can be attributed to the denoising capability of the ResNet. Third, the ZNCCs in the case of “SR-GANw256” appeared somewhat less than those observed in “SR-ResNetw256.” Such a result is not difficult to understand as the SR-GAN tends to rebuild high-frequency details that are different from the original content based on what has been learned in the training stage. Thus, the added details tend to be detrimental to the similarity measure in the pixel space.

It is noted that the GAN merely plays a supporting role to denoise and super-resolve the small-sized watermark extracted from a watermarked color image, presumably after an attack. The purpose of the GAN is to make the recovered watermark more visually recognizable. It is certainly not a problem incorporating the GAN with other watermarking schemes. Although the experimental data were not presented or visualized in detail, our experiences with the other schemes followed a similar trend observed in Figure 7 and Figure 8. Specifically, the ResNet generally made a considerable contribution when the original ZNCC was low; the involvement of the GAN conduced to image clarity perceived by the naked eye but did not necessarily improve the ZNCC. Providing the extracted watermarks were more accurate and precise, the subsequent ResNet and GAN could have the opportunity to enhance the images with the desired resolution and fidelity. It appears that the development of a competent GAN without compromising the ZNCC is also an imperative concern in watermark enhancement.

## 5. Conclusions

We introduced a new type of blind image watermarking scheme that directly embeds a color watermark into a carrier color image. Watermark embedding was implemented by manipulating the magnitudes of paired 2D DCT coefficients, whereas the extraction of the embedded watermark was resolved from the relative positions of these two coefficient magnitudes. To offset the stretching effect on the coefficient magnitudes owing to attacks, we also attached a regulator to restrain the pixel values within a legitimate range.

In a comparison of seven recently developed schemes engaged in high-capacity color-image watermarking, the experimental results prove the superiority of the proposed DCT-MGA scheme in terms of both imperceptibility and robustness. Overall, DCT-MGA makes it possible and reliable to embed a color watermark of size 64×64 into a carrier image of size 512×512 with satisfactory performance. In addition to DCT-MGA, a GAN was employed to enhance the extracted watermark. As demonstrated by the improvement in ZNCC, the generator network renders a color watermark image with a higher resolution and less noise. Nonetheless, the DCT-MGA scheme still lacks sufficient resistance to withstand JPEG compression and median filtering attacks. Developing a numerically embeddable watermarking scheme that survives JPEG compression and median filtering is a topic worthy of further study.

## Figures and Tables

**Figure 1 sensors-23-00370-f001:**
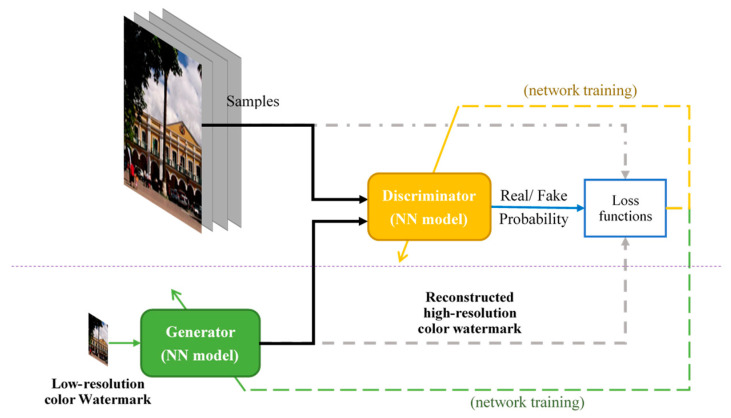
The GAN for denoising and super-resolving recovered watermarks.

**Figure 2 sensors-23-00370-f002:**
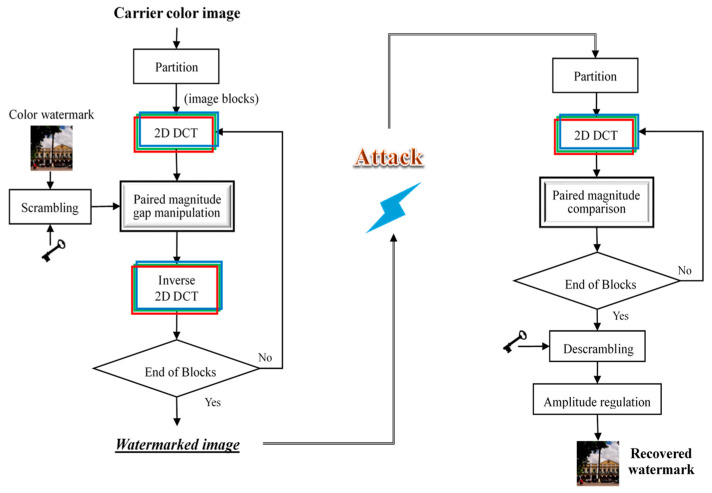
The processing flowchart of the proposed DCT-MGA scheme.

**Figure 3 sensors-23-00370-f003:**
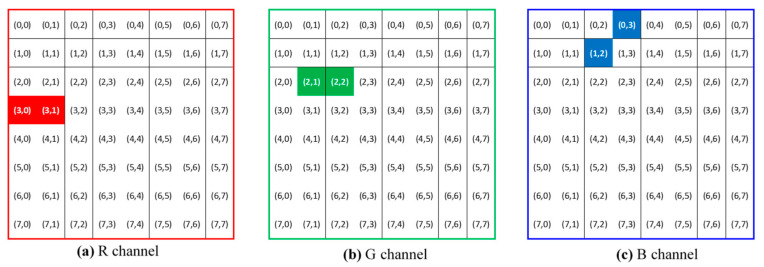
The 2D DCT coefficients chosen for embedding three pixel values (in red, green, and blue colors) in each image block.

**Figure 4 sensors-23-00370-f004:**
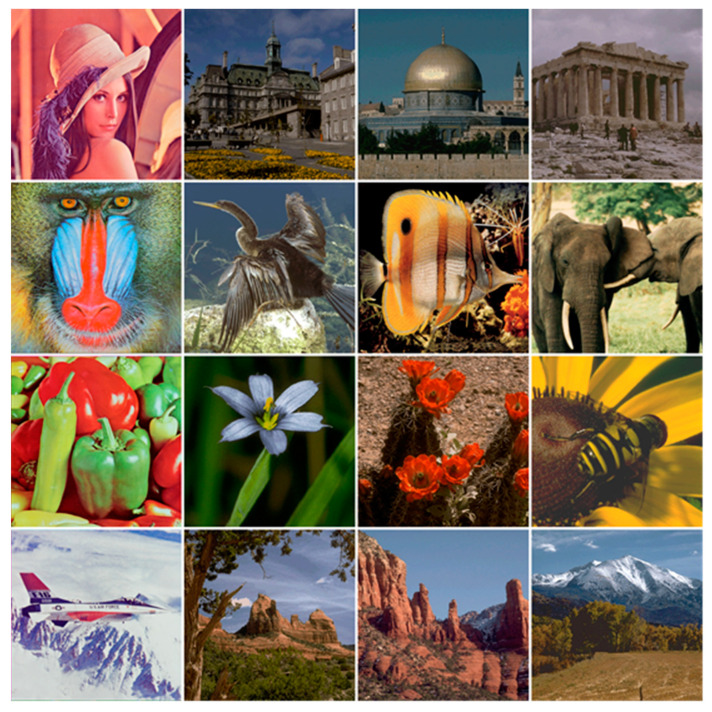
Test color images. Each image comprises 512 × 512 pixels.

**Figure 5 sensors-23-00370-f005:**
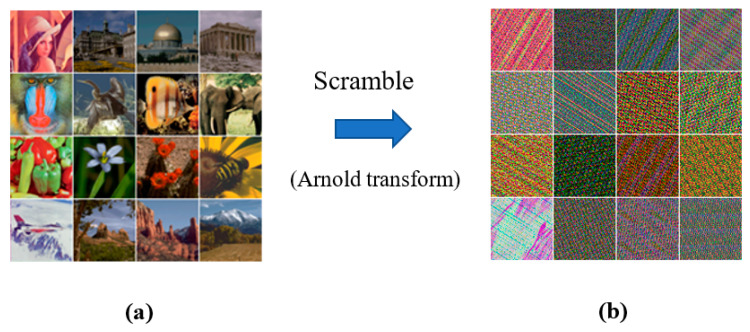
Visual illustration, (**a**) color watermarks, and (**b**) their scrambled versions on the right.

**Figure 6 sensors-23-00370-f006:**
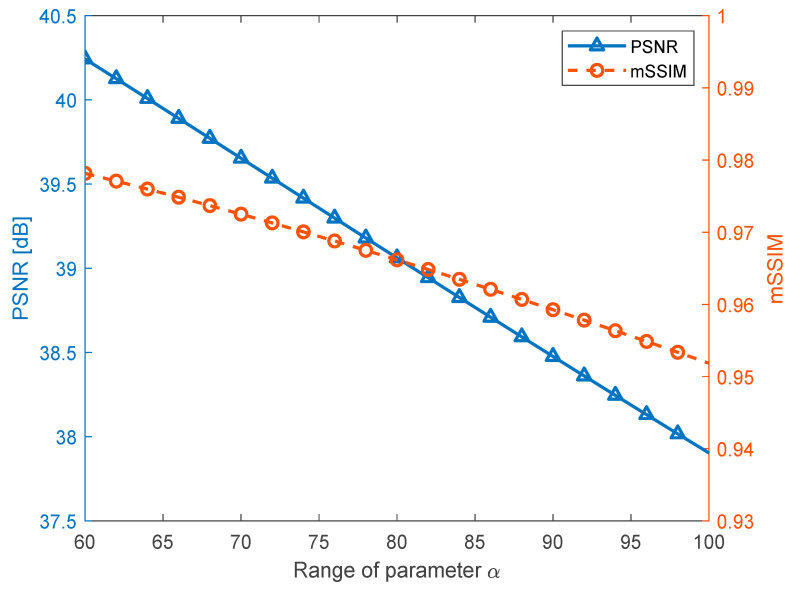
The average PSNRs and mSSIMs resulting from the varying parameter α.

**Figure 7 sensors-23-00370-f007:**
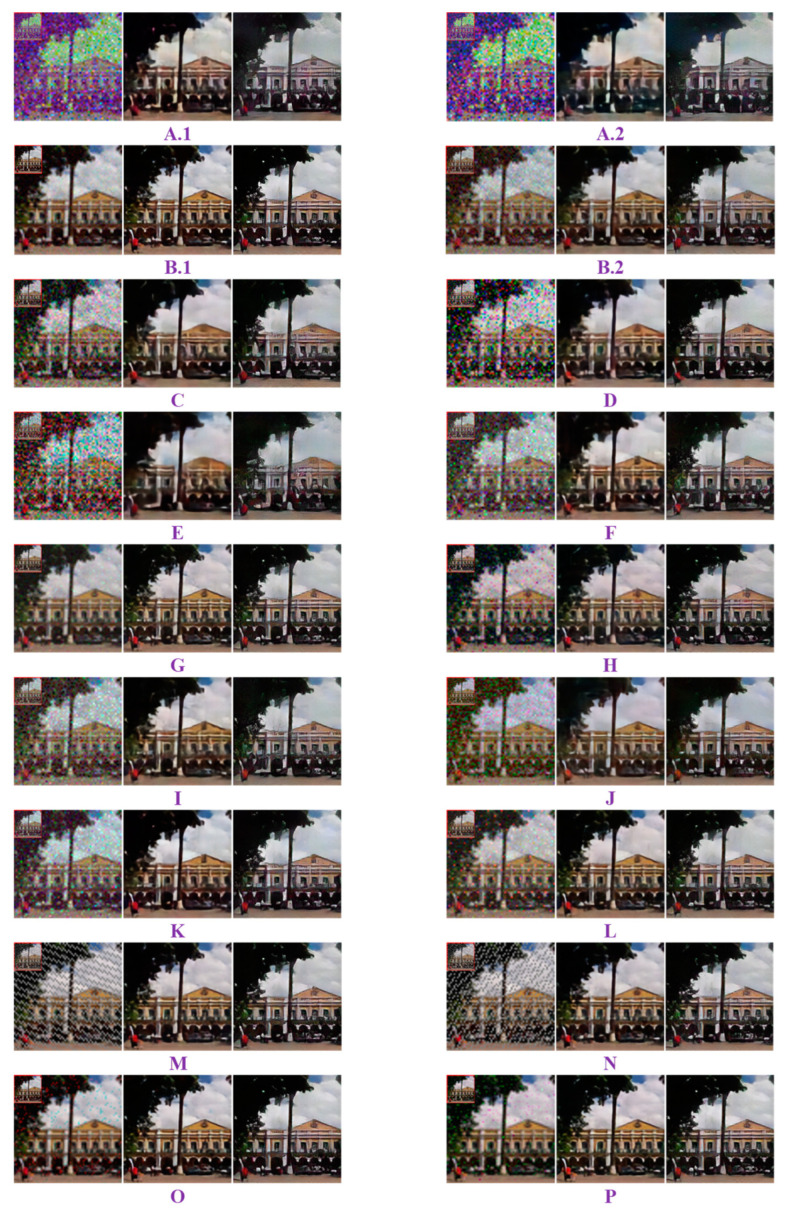
Super-resolved watermarks retrieved from the watermarked “Lena” image under various attacks. The three images from left to right in each cell correspond to the results of “w64/BIw256,” “SR-ResNetw256,” and “SR-GANw256,” respectively. The label underneath each subfigure corresponds to the attack type listed in Table 2.

**Figure 8 sensors-23-00370-f008:**
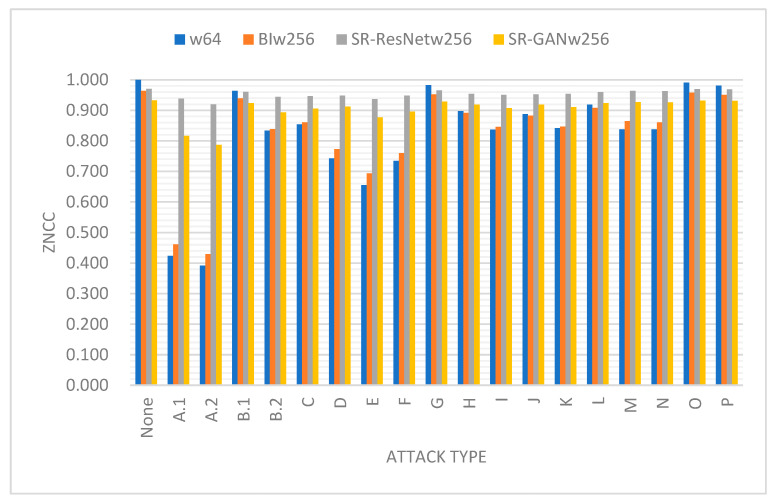
The average ZNCCs computed from four sorts of watermarks.

**Table 1 sensors-23-00370-t001:** The statistics of the measured PSNRs and mSSIMs obtained from compared watermarking schemes. The data in each cell are interpreted as “mean [standard deviation].”

Method	PSNR [dB]	mSSIM
DCT-MGA	38.48 [1.50]	0.959 [0.021]
QRMM22	37.98 [0.67]	0.953 [0.013]
WHT21	34.78 [2.10]	0.958 [0.004]
Haar21	36.86 [0.32]	0.931 [0.028]
Schur21	36.30 [0.17]	0.940 [0.023]
DCT20	32.09 [2.18]	0.944 [0.008]
DWT20	36.13 [0.24]	0.925 [0.026]
DFT20	32.61 [2.29]	0.956 [0.009]

**Table 2 sensors-23-00370-t002:** Attack types with detailed specifications.

Item	Type	Description
**A**	JPEG compression	Apply the JPEG compression to the test image with the quality factor (QF) chosen from {80, 40}.
**B**	JPEG2000 compression	Apply the JPEG2000 compression to the test image with the compression ratio (CR) chosen from {4,8}.
**C**	Gaussian noise corruption	Corrupt the test image using Gaussian noise with the variance set as 0.001 of the full scale.
**D**	Salt-and-pepper noise corruption	Corrupt the test image using the salt-and-pepper noise with 1% intensity.
**E**	Speckle noise corruption	Add the multiplicative noise with a variance of 0.01 to the test image
**F**	Median filtering	Apply a median filter with a 3 × 3 mask to the test image.
**G**	Lowpass filtering	Apply a Gaussian filter with a 3 × 3 mask to the test image.
**H**	Unsharp filtering	Apply an unsharp filter with a 3 × 3 mask to the test image.
**I**	Wiener filtering	Apply a Wiener filter with a 3 × 3mask to the test image.
**J**	Histogram equalization	Enhance the contrast of the test image using histogram equalization.
**K**	Rescaling restoration	Shrink the test image from 512 × 512 to 256 × 256 pixels.
**L**	Rotation restoration	Rotate the test image counterclockwise by 45°.
**M**	Cropping (I)	Crop 25% of the test image on the upper-left corner.
**N**	Cropping (II)	Crop 25% of the test image on the left side.
**O**	Brightening	Add 20 to each pixel value of the test image.
**P**	Darkening	Subtract 20 from each pixel value of the test image.

**Table 3 sensors-23-00370-t003:** The average ZNCCs obtained from the compared watermarking schemes under various attacks.

	Method	DCT-MGA	QRMM22	WHT22	Haar21	Schur21	DCT20	DWT20	DFT20
Attack									
**None**		**1.000**	1.000	0.962	0.958	0.933	1.000	1.000	1.000
**A.1/QF = 80**		**0.334**	0.131	0.192	0.119	0.216	0.190	0.301	0.117
**A.2/QF = 40**		**0.315**	0.039	0.049	0.019	0.074	0.083	0.143	0.061
**B.1/CR = 4**		**0.931**	0.823	0.798	0.659	0.753	0.632	0.919	0.370
**B.2/CR = 8**		**0.749**	0.418	0.413	0.304	0.438	0.330	0.648	0.207
**C**		**0.794**	0.205	0.479	0.021	0.335	0.415	0.535	0.307
**D**		0.669	**0.980**	0.891	0.920	0.757	0.920	0.914	0.901
**E**		**0.701**	0.266	0.473	0.136	0.307	0.445	0.415	0.328
**F**		**0.646**	0.097	0.015	0.065	0.349	0.341	0.460	0.276
**G**		**0.977**	0.724	0.754	0.660	0.769	0.653	0.927	0.422
**H**		**0.869**	−0.035	0.839	−0.018	−0.007	0.754	−0.081	0.649
**I**		**0.793**	0.133	0.086	0.213	0.392	0.335	0.528	0.219
**J**		0.845	0.560	**0.911**	−0.031	0.004	0.867	−0.006	0.576
**K**		**0.792**	0.248	0.110	0.359	0.559	0.396	0.651	0.255
**L**		**0.901**	0.564	0.569	0.458	0.607	0.507	0.776	0.322
**M**		**0.800**	0.760	0.762	0.760	0.732	0.785	0.785	0.785
**N**		**0.798**	0.760	0.758	0.755	0.740	0.786	0.786	0.786
**O**		**0.990**	0.988	0.953	0.304	0.044	0.988	0.985	0.986
**P**		0.913	0.897	**0.917**	0.262	0.053	0.913	0.896	0.898

**Table 4 sensors-23-00370-t004:** The extracted color watermarks from the “Lena” color image for the compared watermarking schemes under various attacks.

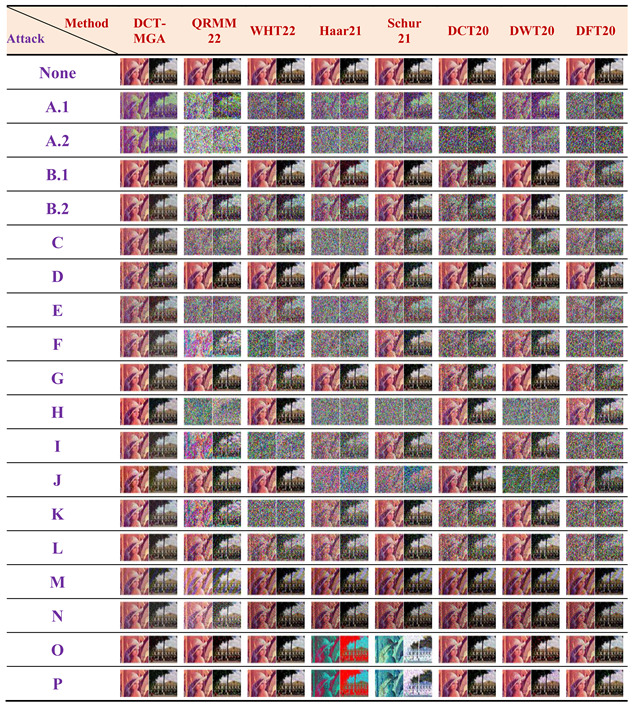

## Data Availability

The image datasets analyzed during the current study are available in the CVG-UGR image database [https://ccia.ugr.es/cvg/dbimagenes/ (accessed on 22 December 2022)], USC-SIPI image database [http://sipi.usc.edu/database (accessed on 22 December 2022)], and IAPR TC-12 Benchmark [https://www.imageclef.org/photodata (accessed on 22 December 2022)]. The watermarking programs implemented in MATLAB code are available upon reasonable request.

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
