# Peer review of "Blind Watermarking for Hiding Color Images in Color Images with Super-Resolution Enhancement"

_sensors, 2022, doi:10.3390/s23010370_

Round 1
Reviewer 1 Report
This paper proposes a new approach to including watermarks in images.
The paper is interesting and well-written. It technically sounds.
A comparison with the state of the art is performed.
My only concern regards the GAN.
It appears orthogonal to the rest of the work and may be applied to other solutions.
Section 4.3 does not include the effect of the GAN on the other papers.
Author Response
Please refer to the attached file for more details.

Reviewer 2 Report
This paper proposes a method of hiding watermarks in carrier images in the discrete cosine transform domain. A large number of experimental results are shown to verify the effectiveness of the proposed algorithm. I recommend the acceptance of this paper. However, there are some problems to be modified:
1. The author should list the PSNR indicators compared with other similar algorithms (other papers).
2. Please describe in detail the advantages and disadvantages of the proposed algorithm.
3. In the experiment section, please give a summary of each table or picture.
4. Some books related to the work of watermarking, the authors may introduce those books.
(1) Feng-Hsing Wang, Jeng-Shyang Pan and Lakhmi C. Jain, Innovations in Digital Watermarking Techniques, Springer, Berlin-Heidelberg, Germany, 2009
(2) Jeng-Shyang Pan, Hsiang-Cheh Huang and Lakhmi C. Jain (editors), Intelligent Watermarking Techniques, World Scientific Publishing Company, Singapore, Feb. 2004
Author Response

(The authors gave the same response as above.)
